# Consequences of *IDH1/2* Mutations in Gliomas and an Assessment of Inhibitors Targeting Mutated IDH Proteins

**DOI:** 10.3390/molecules24050968

**Published:** 2019-03-09

**Authors:** Bozena Kaminska, Bartosz Czapski, Rafal Guzik, Sylwia Katarzyna Król, Bartlomiej Gielniewski

**Affiliations:** 1Laboratory of Molecular Neurobiology, Nencki Institute of Experimental Biology of the Polish Academy of Sciences, 02-093 Warsaw, Poland; bartolomeo.capellini@gmail.com (B.C.); r.guzik@nencki.gov.pl (R.G.); s.krol@nencki.gov.pl (S.K.K.); b.gielniewski@nencki.gov.pl (B.G.); 2Postgraduate School of Molecular Medicine, Medical University of Warsaw, 02-093 Warsaw, Poland; 3Department of Biochemistry, Faculty of Medicine and Health Sciences, Andrzej Frycz Modrzewski Krakow University, 30-705 Krakow, Poland; 4Department of Neurosurgery, Mazovian Bródnowski Hospital, 03-242 Warsaw, Poland

**Keywords:** IDH mutations, metabolic disturbances, epigenetics, gliomas, tumor microenvironment, IHD mutant inhibitors

## Abstract

Isocitrate dehydrogenases (IDH) 1 and 2 are key metabolic enzymes that generate reduced nicotinamide adenine dinucleotide phosphate (NADPH) to maintain a pool of reduced glutathione and peroxiredoxin, and produce α-ketoglutarate, a co-factor of numerous enzymes. *IDH1/2* is mutated in ~70–80% of lower-grade gliomas and the majority of secondary glioblastomas. The mutant IDH1 (R132H), in addition to losing its normal catalytic activity, gains the function of producing the d-(*R*)-2-hydroxyglutarate (2-HG). Overproduction of 2-HG in cancer cells interferes with cellular metabolism and inhibits histone and DNA demethylases, which results in histone and DNA hypermethylation and the blockade of cellular differentiation. We summarize recent findings characterizing molecular mechanisms underlying oncogenic alterations associated with mutated IDH1/2, and their impact on tumor microenvironment and antitumor immunity. Isoform-selective IDH inhibitors which suppress 2-HG production and induce antitumor responses in cells with *IDH1* and *IDH2* mutations were developed and validated in preclinical settings. Inhibitors of mutated IDH1/2 enzymes entered clinical trials and represent a novel drug class for targeted therapy of gliomas. We describe the development of small-molecule compounds and peptide vaccines targeting IDH-mutant gliomas and the results of their testing in preclinical and clinical studies. All those results support the translational potential of strategies targeting gliomas carrying IDH1 mutations.

## 1. Functions of Isocitrate Dehydrogenases

Since the initial discovery of mutations in the isocitrate dehydrogenase 1 (*IDH1*) gene by whole-genome sequencing in a large subset of human gliomas [1], and in acute myelogenous leukemia (AML) [2], much interest was focused on understanding consequences of mutations in *IDH* genes and their roles in tumor progression. Isocitrate dehydrogenases 1 and 2 are key Krebs cycle enzymes that are nicotinamide adenine dinucleotide phosphate (NADP^+^)-dependent and catalyze the oxidative decarboxylation of isocitrate to α-ketoglutarate (α-KG). There are three IDH isoforms, IDH1, IDH2, and IDH3, encoded by different genes. NADP-dependent IDH1 and IDH2 share considerable sequence similarity (70%) and an almost identical protein structure [3]. IDH3 has a unique sequence, is a NAD-dependent enzyme [4], and plays a central role in energy production. To date, there are no reports of tumor-associated mutations in the *IDH3* gene. IDH1 mainly occurs in the cytoplasm and peroxisomes, while IDH2 and IDH3 are found in the mitochondrial matrix [5]. IDH1/2 proteins catalyze the oxidative decarboxylation of isocitrate to α-KG to produce reduced nicotinamide adenine dinucleotide phosphate (NADPH) from NADP^+^. IDH1 and IDH2 enzymes maintain an adequate pool of reduced glutathione (GSH) and peroxiredoxin by providing NADPH. This maintains redox balance, protecting the cell against oxidative damage from various cellular stressors. NADPH generated by IDH1 is involved in lipid metabolism [6] and contributes to the cellular defense against reactive oxygen species (ROS) induced during lipid oxidation [7]. IDH1 and IDH2 participate in protection from oxidative stress [8] by producing molecules such as NADPH and α-KG which have strong reductive properties and protect against DNA damage via their interactions with glutathione- and thioredoxin-producing systems [9]. The reaction driven by IDH1 is the main source of NADPH in the human brain, producing as much as 65% of the brain’s NADPH [10]. IDH1 and IDH2 are also involved in glutamine metabolism under hypoxia and electron transport chain alterations [11]. 

## 2. Pathophysiology of Isocitrate Dehydrogenase Mutations

High-density oligonucleotide arrays and next-generation sequencing of glioma samples of grades II and III (according to classification of the World Health Organization, WHO) revealed an unexpected spectrum of mutations, among which somatic, recurrent mutations in the *IDH1* gene were found in 12% of glioma samples [1]. IDH mutations occur early in pathogenesis of gliomas and persist throughout progression of a glioma from a neural stem or progenitor cell. All known IDH mutations are invariably monoallelic. Mutations in *IDH1* and *IDH2* genes are mostly missense variants leading to a single amino-acid substitution of arginine residues at codon 132 in exon 4 of the *IDH1* gene or codons 140 or 172 of the *IDH2* gene (IDH1-R132, IDH2-R140, or IDH2-R172). IDH1-R132 mutants have dominant-negative, inhibitory effects on wild-type IDH1 (IDH1-wt) in vitro [12].

In addition to losing its catalytic activity, mutant IDH1 and IDH2 enzymes gain the function of catalyzing the reduction of α-ketoglutarate (KG) to its (*R*)-enantiomer of 2-hydroxyglutarate (2-HG) [13]. The 2-HG compound has properties of an oncometabolite, and its accumulation in the cell contributes to cancerogenesis [14]. An oncometabolite is typically a small molecule (or enantiomer), which participates in normal metabolism, but whose accumulation causes metabolic deregulation and, consequently, predisposes cells for future progression to cancer. This term is assigned to *R*(−)-2-hydroxyglutarate ((*R*)-2HG). The 2-HG compound adopts almost identical location to α-KG at the catalytic sites of DNA hydroxylases and enzymes containing the Jumonji domain. 2-HG is most potent against JMJD-containing histone demethylases (JMJD2A, JMJD2C, and FBXL11) with IC_50_ values of approximately 100 μM, suggesting that the JMJD-containing histone demethylases, which includes nearly 30 distinct enzymes in mammalian cells, are probably the main target of 2-HG inhibition. Accumulation of 2-HG in cancer cells results in the complete inhibition of Jumonji-class histone demethylases [15,16]. Changes in histone methylation profiles (especially H3K9 methylation) are associated with *IDH* mutations and result in inhibition of cell differentiation [17]. Moreover, 2-HG is a competitive inhibitor of the ten-eleven translocation (TET) family of 5-methlycytosine hydroxylases responsible for the demethylation of DNA [18,19] (Figure 1). 

A high concentration of 2-HG also promotes angiogenesis via inhibition of prolyl-hydroxylases and stabilization of hypoxia-induced factor (HIF1α), a transcription factor which controls genes promoting cell adaptation to hypoxia, i.e., vascular endothelial growth factor (VEGF) [12]. High HIF1α expression was found in 15% of *IDH*-mutant (IDH-mut) tumors compared to 7.7% of *IDH*-wt tumors [9]. More detailed studies revealed that *IDH* mutation status is associated with a distinct angiogenesis transcriptome signature, decreased expression of HIF1α targets, and impairment of downstream biological functions such as angio- and vasculogenesis that are critical for tumor growth [20].

*IDH* mutations cause profound changes in global cellular metabolism. Initial studies of the effect of *IDH* mutations on the tricarboxylic acid (TCA) cycle function failed to demonstrate significant alterations in TCA cycle metabolites [14]. However, a more detailed study by Reitman et al. [21], who profiled >200 metabolites in *IDH1*- or *IDH2*-mut oligodendroglioma cells, detected changes in levels of amino acids, glutathione metabolites, choline derivatives, and tricarboxylic acid cycle intermediates. These changes mimicked those identified after treatment of the cells with 2-HG. *N*-Acetyl-aspartyl-glutamate (NAAG), a common dipeptide in brain, was 50-fold reduced in *IDH1*-mut expressing cells and 8.3-fold reduced in *IDH2*-mut expressing cells. A similar reduction of NAAG was detected in *IDH*-mut glioma tissues [21]. Acetyl coenzyme A (CoA), which is generated from citrate in the cytoplasm, was shown to regulate the acetylation of cytoplasmic proteins. *IDH*-mut tumors exhibit perturbed acetyl-CoA metabolism and reduced cytosolic acetyl-CoA concentrations, which may result in altered acetylation and activity of many tumorigenic proteins [22]. *IDH1*-mut cells shared multiple metabolic changes with 2-HG-treated cells, suggesting that the oncometabolite production is responsible for the observed metabolic effects [14].

IDH1 activity is also an important factor in metabolic adaptation, which supports an aggressive growth of primary glioblastomas (GBM) maintained despite difficult metabolic conditions. Primary GBMs develop de novo and are the most malignant brain tumors. A wild-type IDH1 mRNA (messenger RNA) and protein are commonly overexpressed in primary GBMs and increased IDH1-wt activity was found in 65% of those tumors. Genetic and pharmacological inactivation of IDH1 decreased GBM cell growth, promoted more differentiated phenotype, increased apoptosis in response to targeted therapies, and prolonged survival of animals with patient-derived xenografts. This forced IDH inactivation/inhibition was accompanied by reduced α-KG and NADPH levels, increased ROS production, enhanced histone methylation, and increased expression of differentiation markers [23]. This suggests that IDH1 upregulation represents a metabolic adaptation of GBM to support growing demands of macromolecular synthesis in tumor cells. 

## 3. Detection of IDH Mutations Improves Classification of Gliomas and Predicts Better Survival 

Gliomas are primary tumors of the central nervous system (CNS) that originate from transformed neural stem or progenitor glial cells, and they were divided by the World Health Organization (WHO) into low-grade gliomas (LGG, WHO grades I and II) and high-grade gliomas (HGG, WHO grades III and IV). LGG are well-differentiated, slow-growing tumors, whereas HGG are less differentiated or anaplastic and diffusive, strongly infiltrating brain parenchyma and making a surgical resection difficult. Histological classification is currently assisted by molecular genetic studies that provide diagnostic, prognostic, and predictive values, and an *IDH* genotype was recently added as the one of the key molecular factors to the classification of gliomas. The new 2016 WHO classification scheme divides diffuse gliomas into LGGs and glioblastomas (GBMs) based on histology. LGGs are further divided into *IDH* wild type or mutant, which is further classified into either an oligodendroglioma that harbors 1p/19q co-deletion or a diffuse astrocytoma that has an intact 1p/19q loci, but is enriched for *ATRX* and *TP53* mutations [24]. 

According to recent WHO classification glioblastomas are divided into an *IDH*-wt GBM, which corresponds to a primary or *de novo* GBM, and an *IDH*-mut GBM, which refers to a secondary or progressive GBM. A mutation in *IDH1* is sufficient to induce genome-wide changes in DNA methylation patterns, including the glioma cytosine phosphate guanine (CpG) island methylator phenotype (G-CIMP) found in a subset of gliomas, which is associated with diverse transcriptional changes [17,25]. G-CIMP is characterized by hypermethylation of CpG islands. Among *IDH*-mut astrocytomas, methylation profile clustering can further subdivide these tumors into G-CIMP–low and G-CIMP–high, reflecting low or high DNA methylation with a significant difference in survival. DNA methylation studies provided an evidence that G-CIMP–high tumors may in fact progress to those that are G-CIMP–low [26]. DNA methylation patterns in *IDH*-mut GBMs are distinct from lower-grade astrocytomas [27]. The identification of clinically relevant subsets of G-CIMP tumors (G-CIMP–high and G-CIMP–low) provided a further refinement in glioma classification that is independent of grade and histology. Many studies showed that patients with *IDH*-mut gliomas have better survival compared to their *IDH*-wt counterparts irrespective of histology and grade, making *IDH* mutation the most important prognostic factor for survival, followed by age, tumor grade, and *O*^6^-methylguanine-DNA methyltransferase gene (*MGMT*) status (reviewed in Reference [28]). Clinical statistics show that a median overall survival (OS) is 31 months for secondary GBM patients with IDH mutations compared to 15 months for those without the mutations. Patients with *IDH*-mut anaplastic astrocytoma have 65 months of median OS compared to 20 months in their *IDH*-wt counterparts [29]. The presence of *IDH1* mutations in anaplastic oligodendroglioma patients is a very strong prognostic factor for OS, but has no a predictive significance for outcome to PCV chemotherapy (adjuvant procarbazine, 1-(2-chloroethyl)-3-cyclohexyl-l-nitrosourea, and vincristine) [30]. All reasons for which glioma patients with *IDH1* mutations show better therapeutic responses and longer survival remain unclear. 

Almost 100% of tumors of oligoastrocytic and oligodendrocytic origin harbor *IDH1/2* mutations; up to now, there is no evidence of any mutations in IDH3 in glial tumors [29]. *IDH1/2* mutations were found in a majority of secondary GBMs (derived from lower-grade tumors) [29], but only 2–3% were found in primary GBMs [31] or pediatric gliomas [32]. *IDH* mutations are considered to be the primary initiating event in WHO grade II/III gliomas and secondary GBMs [33]. 

Several concepts were conceived to explain how *IDH* mutations influence patient outcome. It is believed that decreased MGMT expression, caused by the *MGMT* gene promoter methylation, has a major influence on GBMs responsiveness to alkylating agent therapy (i.e., temozolomide, TMZ) [34]. *IDH* mutations occur more frequently in young patients (younger age of diagnosis) with WHO grade II/III gliomas, who have generally better prognosis [29,35]. However, the *IDH1* mutation does not directly or always correlate with patient survival. Among adult GBM patients who survived at least 36 months, less than one-quarter of them were associated with the *IDH1*-mut status [36]. However, the presence of *IDH1* mutation was a weak prognostic factor in GBM patients with a long-term survival [37]. Also, in the case of low-grade oligodendroglial tumors, the mutation in *IDH1* was not a prognostic factor. While 91% of oligodendrogliomas harbored the *IDH1* mutation, the survival times of patients with *IDH1*-mut tumors were not different compared to patients with *IDH1*-wt tumors. Patients with *IDH1*-mut diffuse astrocytomas lived significantly longer. This suggests that *IDH* mutations could be a prognostic factor for diffuse astrocytoma, but not for oligodendroglioma [38]. 

Several reports pointed out that prognosis for glioma patients with the *IDH1* mutation is associated with DNA methylation patterns. There is a subtype of glioma characterized by the presence of the *IDH* mutation and a low level of DNA methylation (G-CIMP–low) which was associated with a poor outcome [26]. Additionally, the impact of *IDH1* mutations for patient survival may depend on other factors, such as alterations in *1p/19q, ATRX, PTEN*, or *MGMT* methylation status [39,40]. Ki-67 expression in combination of *IDH*-mut may also influence patient survival. Patients with *IDH1/2*-mut and a low level of Ki-67 expression had a relatively good prognosis, while patients with *IDH1/2*-mut and a high level of Ki-67 expression had significantly worse prognosis and shorter times of survival [41]. Outcome of patients with *IDH* mutation was also associated with the occurrence of copy number alterations (CNAs). Patients characterized by the presence of one of the CNAs, +7q, +8q, −9p, or −11p, were associated with worse prognosis and worse overall survival when compared to other patients with the *IDH* mutation [42]. Additionally, prognosis and time of survival of patients with the *IDH* mutation may be gender-dependent; the presence of the *IDH1* mutation was associated with a longer time of survival in male, but not in female patients [43]. 

While the presence of *IDH* mutations commonly correlates with better outcome of glioma patients, some studies showed the connection between the presence of *IDH* mutation and seizure risk in glioma patients. A majority of patients with WHO grade II astrocytoma (but not GBM) suffered pre-operative seizures related to the presence of *IDH* mutation [44]. A recent meta-analysis confirmed that the presence of *IDH1* mutation is correlated with the higher number of preoperative seizures in LGG [45]. 

## 4. Impact of *IDH* Mutations on Glioma Microenvironment

The microenvironment of gliomas is heterogeneous, and there are numerous cancer and non-cancerous, stromal cells which dynamically interact with themselves and with an extracellular matrix. In glioblastoma, this microenvironment includes reactive astrocytes, endothelial cells, and many types of immune cells, among which a main component (up to 30% of a tumor mass) are glioma-associated microglia and macrophages. This population is composed of brain-resident microglia, infiltrating, blood-derived macrophages, and myeloid-derived suppressive cells. Tumor-derived molecules attract and reprogram infiltrating microglia and macrophages and convert them into the cells that support invasion and produce local and systemic immunosuppression (for a review, see References [46,47]). 

Recent studies provided emerging insights into how *IDH* mutations affect the glioma microenvironment. Cytotoxic T lymphocytes (CD8^+^, cluster of differentiation 8 positive) are crucial components of the tumor-specific adaptive immunity. Lymphocyte infiltration occurs to some extent in glioma, and the presence of tumor-infiltrating lymphocytes (TILs) is predictive of clinical outcome [48,49]. The number of CD8^+^ TILs was inversely correlated with tumor grade, whereas the number of CD4^+^ TILs was positively correlated with tumor grade. FoxP3^+^ (forkhead box P3) regulatory lymphocytes were observed only in GBMs [50]. The extent of local glioma-associated CD8^+^ T-cell infiltrate at initial presentation correlates with the long-term survival of GBM patients [51]. The immune checkpoint molecules such as CTLA-4 (cytotoxic T cell antigen 4), PD-1 (Programmed cell death protein 1), PD-L1/2 (Programmed death-ligand 1/2) and others provide inhibitory signals to T cells [52]. In glioma patients, the accumulation of CD4^+^/CD8^+^ T cells and T regulatory cells (Tregs) that express high levels of CTLA-4 and PD-1, or the high expression of PD-L1 in glioma cells correlates with WHO high grade and short survival [53,54]. The impact of *IDH* mutations on immune microenvironment is under debate. Analyses of clinical samples and gene expression data from The Cancer Genome Atlas (TCGA) demonstrated reduced expression of cytotoxic T-lymphocyte-associated genes and interferon (IFN)-γ-inducible chemokines (i.e., *CXCL10*) in *IDH*-mutant tumors compared with *IDH*-wt tumors. Introduction of mutant IDH1 or treatment with 2-HG reduced levels of a chemokine CXCL (C-X-C motif) 10, which was associated with decreased expression of a transcription factor STAT1 (signal transducer and activator of transcription 1), an inducer of inflammation. Forced expression of a mutant *IDH1* also suppressed the accumulation of T cells at tumor sites. Those events were reversed by IDH-C35, a specific inhibitor of a mutant IDH1 [55].

A single study showed that *IDH1* mutation did not associate with increased intratumoral expression of either PD-1+ TIL or PD-L1 in GBMs [56]. However, a recent study demonstrated that *IDH*-wt is associated with the significantly higher TIL infiltration and PD-L1 expression among all grade II–IV gliomas and within the cohort of GBMs [57]. In LGGs and GBMs of TCGA cohorts, significantly higher *PD-L1* gene expression was found in *IDH*-wt compared with *IDH*-mut tumors. Lower *PD-L1* gene expression was associated with the increased promoter methylation in the LGG cohort of TCGA. *IDH*-mut gliomas had higher *PD-L1* gene promoter methylation levels than *IDH*-wt gliomas [57]. PD-L1 expression was significantly associated with a worse clinical outcome in primary and recurrent *IDH*-wt GBMs [58]. As *IDH1*-wt gliomas exhibit increased PD-L1 expression and greater TIL infiltration, those tumors are considered to be more immunologically active and more susceptible to immunomodulatory therapy against PD-1/PD-1L than *IDH*-mut gliomas. Those observations underlie the importance of evaluating *IDH1/2* status in immunomodulatory therapies. 

A flow cytometry analysis of immune composition of human gliomas with a different *IDH1* status demonstrated that human *IDH1*-mut gliomas have significantly lower infiltration of CD45^+^ immune cells, including microglia, macrophages, dendritic cells, B cells, and T cells, compared with *IDH1*-wt gliomas. The downregulated genes in *IDH1*-mut gliomas were associated with immune system processes, and the major Gene Ontology terms were related to chemotaxis and immune cell migration [59]. Introduction of *IDH1*-mut into transgenic mouse gliomas with different genetic background, expressing platelet-derived growth factor alpha (PDGFα), shp53, or Ink4a/Arf^+/+^ and *Ink4a/Arf^+/−^*, demonstrated significantly shorter survival compared to mice with *IDH1*-wt tumors. Similar to human *IDH1*-mut gliomas, reductions in CD45^+^ cells, including microglia, macrophages, monocytes, and polymorphonuclear leukocytes, were reported in the *IDH1*-mut murine tumors. Gene expression in *IDH1*-mut mouse gliomas was negatively associated with leukocyte and neutrophil migration [60]. A computational analysis of relative immune cell content and type of immune response in subtypes of GBMs in the TCGA RNA-sequencing dataset was carried out. All G-CIMP and *IDH1*-mut GBMs were characterized by negative immune responses and lower human leukocyte antigen (HLA) expression [61]. The analyses of complement activation and CD4^+^, CD8^+^, or FoxP3^+^ T-cell infiltration in sections from 72 gliomas of WHO grade III and IV with or without *IDH* mutations showed significantly reduced complement activation and decreased numbers of tumor-infiltrating CD4^+^ and CD8^+^ T cells with comparable FoxP3^+^/CD4^+^ ratios. Ex vivo studies demonstrated that 2-HG inhibits complement activation, decreases cellular C3b(iC3b) opsonization and complement-mediated phagocytosis, and inhibits T-cell migration, proliferation, and cytokine secretion. This is consistent with reduced host immune responses in *IHD*-mut gliomas [62].

A direct link between an *IDH* mutation and T-cell functions was recently demonstrated. It was shown that tumor cell-derived (*R*)-2-HG is taken up by T cells, where it induces perturbation of NFAT (nuclear factor of activated T cells) transcriptional activity and polyamine biosynthesis, which results in suppression of T-cell proliferation and activity. *IDH1*-mut gliomas showed reduced T-cell numbers and altered calcium signaling. Consistently, antitumor immunity to experimental syngeneic *IDH1*-mut gliomas induced by an *IDH1*-specific vaccine or checkpoint inhibition was significantly improved by inhibition of the enzymatic function of mutant IDH1 [59].

Blood vessels in glioblastoma are abnormal and display a distinct gene expression signature compared with vessels in normal brain [63,64]. Those vascular abnormalities are connected to a high expression of angiogenic factors, including vascular endothelial growth factor (VEGF), transforming growth factor (TGF) β2, and pleiotrophin (PTN). In addition to inducing epigenetic deregulation, (*R*)-2HG can regulate the activity of α-KG-dependent dioxygenases, specifically EGLN (EGL nine homolog 1) prolyl 4-hydroxylases [65], that are responsible for regulation of HIF1α, which controls angiogenesis. In *IDH*-mut gliomas (*R*)-2HG acts as an activator of EGLN prolyl 4-hydroxylases, leading to decreased levels of HIF1α and reduced expression of genes implicated in hypoxia, and vasculo- and angiogenesis-related signaling such as: *VEGFA*, *PDGF* (platelet derived growth factor), or *ANGPT2* (angiopoietin-2) [65]. Transcriptomic studies showed that *IDH*-wt LGGs presented a specific angiogenic gene expression signature, including upregulation of *ANGPT2* and *SERPINH1* (SERPIN family H), linked to enhanced endothelial cell migration and matrix remodeling, suggesting that these tumors are more angiogenic than *IDH*-mut LGGs. Transcription factor analysis indicated increased TGFβ and hypoxia signaling in *IDH*-wt LGGs. As a consequence, *IDH*-wt LGG vessels are molecularly distinct from the vasculature of *IDH*-mut LGGs [66]. All reported data indicated gross differences in composition and functionality of different cells creating the microenvironment of *IDH*-wt and *IDH*-mut gliomas. 

## 5. Targeting of Mutant IDH1/2 Gliomas with Isoform-Specific Chemical Inhibitors 

Many preclinical and clinical data validated IDH1/2 as an important target for antitumor drug development. A growing number of studies using cellular and animal models indicate that pharmacological inhibition of mutated IDH1/2 offers therapeutic benefits and there is a rationale for development of isoform-specific inhibitors (Figure 1). In principle, small molecules are designed to bind within the active catalytic site of a mutant *IDH1/2* and block the conformational change required for the enzyme to convert α-KG to 2-HG [67]. Consequently, targeted inhibition of mutated *IDH1/2* results in decreased intracellular and serum levels of 2-HG [68,69], followed by reversion of global alterations in an epigenome. Current targeted inhibitors of *IDH1* (AG120, IDH305), *IDH2* (AG221), and pan-*IDH1/2* (AG881) selectively inhibit the mutant IDH activity and induce cell differentiation in in vitro and in vivo models. Preliminary results from phase I clinical trials with IDH inhibitors in patients with advanced hematologic malignancies demonstrated an objective response rate ranging from 31% to 40% with durable responses (>1 year) [70]. We briefly summarize the properties of IDH1/2 inhibitors below. 

AG-120 is a first-in-class, orally administered, reversible, and highly selective small-molecule inhibitor of mutant IDH1/R132H with half maximal inhibitory concentration (IC_50_) = 40–50 nM. AG-120 reduced intracellular levels of 2-HG, inhibited cell proliferation, and released a block of erythropoietin-induced differentiation in human erythroleukemia TF-1 cells harboring IDH1/R132H in vitro and in primary human blast cells cultured ex vivo [68]. AG-120 is currently being evaluated in several clinical trials for the therapy of patients with relapsed or refractory AML, myelodysplastic syndrome, and advanced solid tumors including glioma, chondrosarcoma, and cholangiocarcinoma with a mutant IDH1/R132H (Table 1). Early results from patients with relapsed or refractory AML indicate that monotherapy is well tolerated at doses ranging from 100 to 1200 mg, and a maximum tolerated dose was not reached. An overall response rate of 36% and a complete response rate of 29.5% were induced [71]. Moreover, dose-escalation clinical trials indicated an overall response rate of 50% [72]. The plasma 2-HG level in patients with *IDH1*-mutant AML was reduced almost completely after AG-120 treatment. Preliminary data from a phase I study of patients with *IDH1*-mut gliomas demonstrated that AG-120 treatment had no dose-limiting toxicity or serious adverse effects [73].

AGI-5198 is a small-molecule, highly selective IDH1/R132H inhibitor with a half maximal inhibitory concentration IC_50_ = 70 nM and IC_50_ > 100 μM for wild-type IDH1 and IDH2 isoforms. It suppressed the production of 2-HG in a dose-dependent manner and significantly inhibited growth of anaplastic oligodendroglioma cells harboring heterozygous IDH1/R132H mutation without influence on IDH1-wt patient-derived glioma cells. AGI-5198 also reduced growth of human IDH1/R132H glioma xenografts in mice and did not impair glioma expressing wild type IDH1, without significant toxicity. AGI-5198 treatment of mice engrafted with *IDH1*-mutant glioma removed repressive marks via significant reduction of H3K9 and H3K27 trimethylation at the promoters of genes associated with gliogenic (astrocytic and oligodendrocytic) differentiation, increasing the expression of these genes [74]. AGI-5198 also reduced 2-HG levels in human chondrosarcoma cells that harbor *IDH1*/R132G and *IDH1*/R132C mutations in a dose-dependent manner. Moreover, AGI-5198 significantly inhibited colony formation and migration of chondrosarcoma cells, without influence on *IDH1*-wt human normal chondrocytes, and induced an apoptotic cell death and G_2_/M cell-cycle arrest in human chondrosarcoma cells in vitro [75]. AGI-5198 is yet to enter clinical trials.

BAY-1436032 is a small-molecule, oral inhibitor of pan-mutant IDH1 with IC_50_ = 3–16 nM. It blocks 2-HG production and induces myeloid differentiation manifested by morphological changes and upregulated expression of CD14 and CD15 markers in patient-derived *IDH1*-mut AML cells cultured ex vivo. In mice with implanted AML xenografts, BAY-1436032 decreased the level of 2-HG in serum nearly to the level found in normal tissues, and promoted differentiation of leukemic blast cells which correlated with the prolonged survival. The pan-mutant IDH1 inhibitor affected leukemia stem cells’ ability to self-renew with downregulation of stemness-associated genes and upregulation of those associated with myeloid differentiation. Similarly to other IDH-mutant inhibitors, BAY-1436032 affected global histone and DNA methylation levels [76]. The inhibitor entered clinical evaluation for the treatment of patients with advanced solid tumors, including anaplastic glioma, GBM, and intrahepatic cholangiocarcinoma.

FT-2102 is a small-molecule, oral allosteric inhibitor of mutant IDH1 (IC_50_ = 10 nM) currently undergoing a clinical study for the treatment of patients with *IDH1*-mut AML or higher-risk myelodysplastic syndrome, who relapsed or are refractory to a prior therapy or were disqualified for standard treatment. It is used at a dose of 150 mg/daily as monotherapy and in combination with azacitidine [67].

HMS-101 is a small-molecule inhibitor of mutant IDH1 which reduced 2-HG level, affected proliferation and the ERK (extracellular signal-regulated kinase) signaling pathway, and inhibited colony formation of *IDH1*-mut murine cells and primary AML cells cultured ex vivo without affecting normal bone marrow cells [77]. In a mouse model of leukemia, HMS-101 blocked the production of 2-HG and inhibited proliferation of *IDH1*-mut cells. It induced cell differentiation, which was correlated with the prolonged survival of mice with *IDH1*-mut AML cells [78]. HMS-101 is yet to be investigated in patients.

IDH305 is a small-molecule, oral, highly selective, allosteric inhibitor of the mutant IDH1/R132H. IDH305 inhibited 2-HG production and tumor cell proliferation with an IC_50_ = 24 nM, and showed an anticancer activity in *IDH1*/132H–mut cells in preclinical studies [79]. IDH305 is under clinical evaluation as a single agent or in combination with standard treatments for the therapy of patients with progressive II or III gliomas, low-grade gliomas with measurable 2-HG levels, AML, and other advanced malignancies harboring *IDH1*/R132H mutations (see summary in Table 1). Early results from clinical studies demonstrated an encouraging anticancer potential of IDH305 and a favorable safety profile in patients with AML harboring mutant IDH1/132H [79].

AGI-6780 is a small-molecule, allosteric inhibitor designed to target the IDH2/R140Q with IC_50_ = 170 nM and IC_50_ > 100 μM for IDH1/132H. It decreased the level of intracellular and extracellular 2-HG in a dose- and time-dependent manner and induced differentiation of IDH2-mut TF-1 human erythroleukemia cells in vitro and primary human AML cells cultured ex vivo [80]. AGI-6780 reversed DNA hypermethylation within several weeks, whereas histone hypermethylation was removed within several days [81]. AGI-6780 is yet to enter clinical trials.

AG-221 (enasidenib, CC-90007) is an orally available, reversible, and highly selective inhibitor of the mutant IDH2/R140Q with IC_50_ = 12 nM. AG-221 reduced serum levels of 2-HG and induced myeloid differentiation of AML leukemic blast cells engrafted to immunodeficient mice. AG-221 is currently being evaluated in several clinical trials for the use in advanced hematologic malignancies positive for a mutated IDH2 [69]. AG-221 decreased 2-HG level in marrow, plasma, and urine of xenotransplant mice, and promoted significant survival benefits in a dose-dependent manner [82]. At present, AG-221 is being investigated in clinical trials for the therapy of patients with advanced *IDH2*-mut hematologic malignancies. Preliminary clinical data indicate that monotherapy with AG-221 resulted in up to a 98% decrease in 2-HG level in plasma, and the drug was well tolerated [83]. Levels of 2-HG in patients with IDH2-mut AML were lowered to the level in healthy volunteers [84]. AG-221 treatment promoted differentiation of leukemic cells into mature myeloid cells [72], and induced an objective overall response rate of 41% and a complete response in 28% patients with AML [72]. An overall response rate in patients with other hematological malignancies was 56% [72]. Moreover, AG-221 was clinically developed for treatment in combination with standard chemotherapy, and hypomethylating agents in newly diagnosed AML patients [70]. Recently, the inhibitor entered clinical trials for the treatment of patients with advanced solid tumors including glioma, chondrosarcoma, and cholangiocarcinoma with a mutated *IDH2* (Table 1). 

AG-881 is a small-molecule, orally administered pan-IDH1/2-mutant inhibitor. Preclinical studies indicated that AG-881 blocks both mutated IDH1 and IDH2 proteins with IC_50_ = 0.04–22 nM, decreases the level of 2-HG, and crosses the blood–brain barrier. AG-881 was shown as a full brain-penetrant and, thus, may possibly represent a more effective agent for the therapy of patients with *IDH1/2*-mut gliomas [67]. A pan-IDH1/2 inhibitor is suggested to be a second-generation drug in *IDH*-mut cancers [70]. AG-881 is currently being investigated in clinical trials for patients with solid tumors, including gliomas and advanced hematologic malignancies harboring mutated *IDH1* and/or *IDH2* that progressed prior to treatment with the use of mutant IDH inhibitors [67,70]. As AG-881 recently entered the clinical evaluation in patients, no data reports are yet available (Table 1).

## 6. Development of IDH1-R132H Targeting Peptide Vaccines 

It was demonstrated that an IDH1-R132H protein contains an immunogenic epitope suitable for development of a mutant protein-specific vaccine. A fraction of *IDH1*-mut glioma patients have isoform-specific antibodies and displayed an IFN-γ T-cell response against a mutant IDH1 (mutIDH1) protein. Peptides encompassing the altered region were presented on major histocompatibility complexes (MHC) class II and induced specific CD4^+^ T-helper-1 (TH1) responses against an altered IDH1. Screening of peptide libraries around the altered region of the IDH1-R132H was performed to identify peptides that would induce an interferon-γ (IFN-γ) responses in T cells. A peptide vaccine consisting of a 20-mer peptide was derived from the IDH1-R132H. Peptide vaccination of humanized mice (transgenic for human MHCI and II molecules) with IDH1-R132H tumors resulted in induction of specific antitumor immune responses and restriction of growth of syngeneic IDH1-R132H-expressing tumors. T-cell depletion abrogated the reduction in *IDH1*-mut tumor growth after IDH1 peptide vaccination [86]. A similar good efficacy of mutIDH1 vaccine was demonstrated in a murine GL261 model, in which immunization of mice bearing mutIDH1/R132H GL261 gliomas was carried out. Mice with mIDH1-GL261, but not parental GL261 gliomas, survived longer than controls when treated with mutIDH1 peptides; 25% of them were cured. Vaccination with peptides resulted in higher counts of peripheral CD8^+^ T cells, higher levels of IFN-γ, and the presence of anti-mIDH1 antibodies. Intratumoral upregulation of IFN-γ, Granzyme-B, and Perforin-1, together with downregulation of TGFβ2 and IL (interleukin)-10 suggested rising antitumor immunity [87]. It was followed by a clinical trial (NOA-16) of the IDH1 peptide vaccine targeting the IDH1-R132H to evaluate the safety and tolerability, as well as immune responses to the vaccine in patients having IDH1-R132H malignant gliomas (https://clinicaltrials.gov/ct2/show/NCT02454634). First reported results showed that NOA-16 demonstrated safety and immunogenicity of a mutant IDH1-R132H peptide vaccine in patients with newly diagnosed IDH1-R132H malignant astrocytomas [88]. These encouraging results provided a strong evidence that a mutation-specific IDH1-R132H vaccine may represent a viable novel therapeutic strategy for IDH1-R132H-mutant tumors [89].

## 7. Conclusions

Several studies provided strong evidence for the oncogenic potential of *IDH1/2* mutations, leading to the production of an oncometabolite 2-HG, which alters epigenetic regulation, cancer cell differentiation, and cell metabolism. Depending on associated genomic aberrations and a cellular context, the oncogenic potential of *IDH1/2* mutations ranges from an initiating event, which promotes transformation, to a secondary oncogenic event conferring selective advantage to cancer cells. In vitro and in vivo preclinical studies demonstrated that inhibition of mutated IDH1/2 enzymes reduces intracellular 2-HG levels, reverses epigenetic deregulation, and releases the differentiation block in cancer cells. These findings provided a rationale for initiation of preclinical and a few clinical trials evaluating novel, isoform-specific, mutated IDH1/2 inhibitors in cancers with such genomic alteration. Novel inhibitors of mutant IDH1 (AG120, IDH305), IDH2 (AG221), and pan-IDH1/2 (AG881) were developed that selectively inhibit mutant IDH proteins and induce cell differentiation in in vitro and in vivo models. Preliminary results from phase I clinical trials with those inhibitors demonstrated a response rate ranging from 31% to 40% with durable responses (>1 year) in patients with advanced hematologic malignancies and a positive activity in solid tumors with *IDH* mutations, such as cholangiocarcinomas and low-grade gliomas. The mutated IDH1-R132H vaccines were developed and proven to be effective in launching antitumor immunity in preclinical models, which led to initiation of a clinical trial in glioma patients.

Current clinical trials evaluating potential inhibitors in cancers with mutant *IDH1/2* are aimed at confirming their safety and tolerability profiles, and clinical activity as a single agent or in combination with standard treatment strategies. AG-120 and AG-221 obtained fast track and orphan drug designations from the United States Food and Drug Administration (FDA). Preliminary results from ongoing clinical trials indicate that pharmacological, small-molecule inhibitors of mutant IDH1/2 have promising activity and efficacy in patients with relapsed and/or refractory cancer disease, as discussed in an elegant recent review [90]. Generally, the treatment was relatively well tolerated and no maximal tolerated doses were reached and those inhibitors showed less toxic adverse effects than standard chemotherapeutic drugs [90]. 

## Figures and Tables

**Figure 1 molecules-24-00968-f001:**
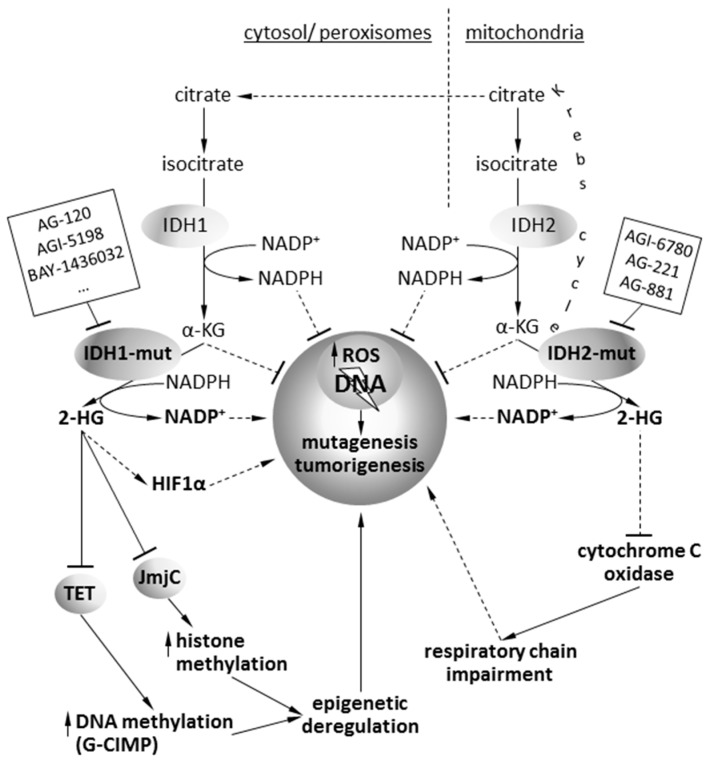
Summary of metabolic and epigenetic alterations induced by *IDH* mutations in cancer cells. The action mode of inhibitors targeting the mutant isocitrate dehydrogenase (IDH) proteins is indicated.

**Table 1 molecules-24-00968-t001:** A summary of clinical trials with isoform-specific isocitrate dehydrogenase inhibitors.

Inhibitor	Target	Cancer	Current Status of Clinical Trials	Identifier at ClinicalTrials.Gov	Company	Reference
AG-120	Mutant IDH1	Cholangiocarcinoma, chondrosarcoma, glioma and advanced solid tumors	Phase I	NCT02073994	Agios Pharmaceuticals Inc./Celgene Corporation	[71,73,85]
		Advanced hematologic malignancies: relapsed or refractory AML, untreated AML, other hematologic malignancies	Phase I	NCT02074839	Agios Pharmaceuticals Inc./Celgene Corporation	
		Newly diagnosed AML, untreated AML, AML arising from MDS, AML arising from antecedent hematologic disorder, AML arising after exposure to genotoxic injury	Phase I	NCT02632708	Agios Pharmaceuticals Inc./Celgene Corporation	
		Newly diagnosed AML	Phase Ib/II	NCT02677922	Agios Pharmaceuticals Inc./Celgene Corporation	
AG-221 (Enasidenib)	Mutant IDH2	Advanced hematologic malignancies	Phase I/II	NCT01915498	Agios Pharmaceuticals Inc./Celgene Corporation	[72,82,83,84,85]
		Advanced solid tumors including glioma, angioimmunoblastic T-cell lymphoma, intrahepatic cholangiocarcinoma chondrosarcoma	Phase I/II	NCT02273739	Agios Pharmaceuticals Inc./Celgene Corporation	
		Late-stage AML	Phase III	NCT02577406	Agios Pharmaceuticals Inc./Celgene Corporation	
		Newly diagnosed AML, untreated AML, AML arising from MDS, AML arising from AHD, AML arising after exposure to genotoxic injury	Phase I	NCT02632708	Agios Pharmaceuticals Inc./Celgene Corporation	
		Newly diagnosed AML	Phase Ib/II	NCT02677922	Agios Pharmaceuticals Inc./Celgene Corporation	
AG-881	Mutant IDH1 and IDH2	Advanced hematologic malignancies: AML, MDS	Phase I	NCT02492737	Agios Pharmaceuticals Inc./Celgene Corporation	[67,70]
		Advanced solid tumors: cholangiocarcinoma chondrosarcoma, gliomas	Phase I	NCT02481154	Agios Pharmaceuticals Inc./Celgene Corporation	
AGI-6780	Mutant IDH2	AML	-	-	Agios Pharmaceuticals Inc.	[80,81]
AGI-5198	Mutant IDH1	Chondrosarcoma, low-grade WHO glioma	-	-	Xcess Biosciences Inc.	[74,75]
BAY-1436032	Mutant IDH1	Advanced solid tumors, including anaplastic glioma, glioblastoma, intrahepatic cholangiocarcinoma	Phase I	NCT02746081	Bayer	[76]
FT-2102	Mutant IDH1	AML, high-risk MDS	Phase I/Ib	NCT02719574	Forma Therapeutics Inc.	[67]
HMS-101	Mutant IDH1	AML	-	-	Ascenion GmnH	[78]
IDH305	Mutant IDH1	II or III WHO glioma	Phase II	NCT02977689	Novartis AG Pharmaceuticals	[79]
		Low-grade glioma	Phase II	NCT02987010		
		AML and advanced solid tumors including cholangiocarcinoma and glioma	Phase I	NCT02381886		
		AML	Phase I	NCT02826642		

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
