# Peer review of "Consequences of IDH1/2 Mutations in Gliomas and an Assessment of Inhibitors Targeting Mutated IDH Proteins"

_molecules, 2019, doi:10.3390/molecules24050968_

Round 1

Reviewer 1 Report

In this manuscript the Authors review the role of IDH mutations in cancer. They offer a compelling historical background of landmark studies, very complete update on latest developments especially in terms of drug development. Of course, the Authors are commended for their effort and clarity.

I had some very minor observations:

Row 66: the term "oncometabolite" is appropriate, but not described before. The concept is still quite new and I would recommend a quick introduction.

Row 93-96: reference misses for the described findings

Row 98: Authors cite a review. I would recommend original papers.

Row 103: space not needed between "65" and "%"

Row 106-108: This paragraph is a little confusing. It reads like all the listed changes may have a role in the tumorigenic phenotype of IDH mutations (that is instead largely if not entirely ascribable to 2-HG, which is actually not listed). Maybe a slight rephrasing may help the reading.

Chapter 2: an important point to make is reversibility of 2-HG-driven cell transformation (it's only a suggestion, Authors needn't feel obliged to address this)

Row 148: two references are listed separately

Row 192: there is a "by" missing between "composed" and "brain"

Figure 1 has very poor quality and is not readable. Might be a formatting issue.

Author Response

Row 66: the term "oncometabolite" is appropriate, but not described before. The concept is still quite new and I would recommend a quick introduction.

Ad. We added: An oncometabolite is a small molecule (or enantiomer), which participates in  normal metabolism, but whose accumulation causes metabolic deregulation and consequently predisposes cells for future progression to cancer. This term has been assigned to R(-)-2-hydroxyglutarate ((R)-2HG).

Row 93-96: reference misses for the described findings.

Ad. We introduced  the reference to the paper which nicely covers this area. [22] Parker, S.J., Metallo, C.M. Metabolic consequences of oncogenic IDH mutations. Pharmacol. Ther 2015, 152:54-62, doi: 10.1016/j.pharmthera.2015.05.003.

Row 98: Authors cite a review. I would recommend original papers.

Ad. We introduced  the reference to the original paper [21] and a review [22].

Row 103: space not needed between "65" and "%". Corrected.

Row 106-108: This paragraph is a little confusing. It reads like all the listed changes may have a role in the tumorigenic phenotype of IDH mutations (that is instead largely if not entirely ascribable to 2-HG, which is actually not listed). Maybe a slight rephrasing may help the reading.

Ad. In that paragraph we discuss findings showing overexpression of IDH1 in GBM and consequences of IDH1 knockdown or downregulation. It is to show that rapidly proliferating GBMs upregulate IDH1 expression as a part of adaptation mechanism and this is independent on the IDH1/2 mutation. We would like to mention another mechanism which besides 2-HG may contribute to glioma pathogenesis.

Chapter 2: an important point to make is reversibility of 2-HG-driven cell transformation (it's only a suggestion, Authors needn't feel obliged to address this).

Ad. We agree that it is an important issue but we cover this issue in the paragraph where we describe action of inhibitors (see 259): In consequence, targeted inhibition of mutated IDH1/2 results in a decrease of intracellular and serum levels of 2-HG [67,68] followed by reversion of global alterations in an epigenome.

Row 148: two references are listed separately

Ad. It has been corrected.

Row 192: there is a "by" missing between "composed" and "brain"

Ad. It has been corrected.

Figure 1 has very poor quality and is not readable. Might be a formatting issue.

Ad. We provided a new figure of better quality.

Reviewer 2 Report

The review titled. ‘Consequences of IDH1/2 mutations in gliomas and an assessment of inhibitors trageting mutated IDH proteins’ by bozena Kaminska et al. is a well written, broad review. This review was delightful to read. There are some topics and manuscripts I would like to see incorporated into review to make it comprehensive.

1. Description of entire tumor microenvironment in gliomas, including mesenchymal stem cells, pericytes etc. and therapeutic effect of targeting IDH1/2 on these populations of cells in tumor. (some of references below included)

2. The following references should be considered to be included in review.

·         Mutant IDH1 Disrupts the Mouse Subventricular Zone and Alters Brain Tumor Progression. Christopher J. Pirozzi , Austin B. Carpenter , Matthew S. Waitkus, Catherine Y. Wang , Huishan Zhu, Landon J. Hansen , Lee H. Chen , Paula K. Greer , Jie Feng, Yu Wang , Cheryl B. Bock , Ping Fan , Ivan Spasojevic , Roger E. McLendon , Darell D. Bigner , Yiping He,  Hai Yan.

·         Mutant IDH1 regulates the tumor-associated immune system in gliomas. Nduka M. Amankulor, Youngmi Kim, Sonali Arora, Julia Kargl, Frank Szulzewsky, Mark Hanke, Daciana H. Margineantu, Aparna Rao, Hamid Bolouri.

·         IDH1-R132H acts as a tumor suppressor in glioma via epigenetic up-regulation of the DNA damage response. Felipe J. Núñez, Flor M. Mendez , Padma Kadiyala, Mahmoud S. Alghamri, Masha G. Savelieff, Maria B. Garcia-Fabiani, Santiago Haase, Carl Koschmann , Anda-Alexandra Calinescu , Neha Kamran, Meghna Saxena, Rohin Patel , Stephen Carney , Marissa Z. Guo , Marta Edwards , Mats Ljungman, Tingting Qin, Maureen A. Sartor , Rebecca Tagett6 , Sriram Venneti , Jacqueline Brosnan-Cashman , Alan Meeker , Vera Gorbunova , Lili Zhao, Daniel M. Kremer Li Zhang, Costas A. Lyssiotis, Lindsey Jones, Cameron J. Herting, James L. Ross, Dolores Hambardzumy.

·         D-2-Hydroxyglutarate Is an Intercellular Mediator in IDH-Mutant Gliomas Inhibiting Complement and T Cells. Lingjun Zhang , Mia D. Sorensen, Bjarne W. Kristensen, Guido Reifenberger, Thomas M. McIntyre , and Feng Lin. - complement

·         Computational Characterization of Suppressive Immune Microenvironments in Glioblastoma. Suvi Luoto , Isma€l Hermelo , Elisa M. Vuorinen , Paavo Hannus , Juha Kesseli , Matti Nykter, and Kirsi J. Granberg.

·         Integrated genomic characterization of IDH1-mutant glioma malignant progression Hanwen Bai, Akdes Serin Harmanci, E Zeynep Erson-Omay, Jie Li, Süleyman CoÅŸkun, Matthias Simon, Boris Krischek, Koray Özduman, S Bülent Omay, Eric A Sorensen, Åževin Turcan, Mehmet Bakırcığlu, Geneive Carrión-Grant, Phillip B Murray, Victoria E Clark1,, A Gulhan Ercan-Sencicek, James Knight, Leman Sencar, Selin Altınok, Leon D Kaulen, Burcu Gülez, Marco Timmer, Johannes Schramm, Ketu Mishra-Gorur, Octavian Henegariu, Jennifer Moliterno, Angeliki Louvi, Timothy A Chan, Stacey L Tannheimer8, M Necmettin Pamir, Alexander O Vortmeyer, Kaya Bilguvar, Katsuhito Yasuno, and Murat Günel.

CSC/Perictyes –targeting in TME

·         Glioblastoma Stem Cells Generate Vascular Pericytes to Support Vessel Function and Tumor Growth. Lin Cheng,Zhi HuangWenchao ZhouQiulian WuShannon DonnolaJames K. Liu, Xiaoguang Fang,Andrew E. SloanYubin MaoJustin D. LathiaWang MinRoger E. McLendonJeremy N. Rich, and Shideng Bao.

·         Targeting Glioma Stem Cell-derived Pericytes Disrupts the Blood-Tumor Barrier and Improves Chemotherapeutic Efficacy. Wenchao ZhouCong ChenYu ShiQiulian WuRyan C. GimpleXiaoguang FangZhi HuangKui Zhai,1Susan Q. Ke, Yi-Fang PingHua FengJeremy N. RichJennifer S. YuShideng Bao, andXiu-Wu Bian.

3. The description of small molecules targeting IDH1/2 should be standardized. i.e. one paragraph describes IC50 values but no others do and reader could compare for self if information standardized. All in vitro findings should be described if available.

4. Make sure spaces between each paragraph.

Author Response

2. The following references should be considered to be included in review.

Ad. We included additional description and most of the proposed references:

A flow cytometry analysis of immune composition of human gliomas with a different IDH1 status demonstrated that IDH1-mut human gliomas have significantly lower infiltration of  CD45+ immune cells, including microglia, macrophages, dendritic cells, B cells, and T cells, compared with IDH1-wt gliomas. The down-regulated genes in IDH1-mut gliomas were associated with immune system processes and Gene Ontology (GO) terms were related to chemotaxis and immune cell migration [61]. Introduction of IDH1-mut into transgenic mouse gliomas with different genetic background: expressing PDGFα (Platelet derived growth factor alpha), shp53 or Ink4a/Arf+/+ and Ink4a/Arf+/− demonstrated significantly shorter survival compared with mice with IDH1-wt tumors with mice with muIDH1 tumors. Similar to human IDH1-mut gliomas, reductions in CD45+ cells, including microglia, macrophages, monocytes, and polymorphonuclear leukocytes were reported in the IDH1-mut tumors. Gene expression in IDH1-mut mouse gliomas was negatively associated with leukocyte and neutrophil migration [61]. doi: 10.1101/gad.294991.116. A computational analysis of relative immune cell content and type of immune response in subtypes of GBMs from TCGA RNA-seq data set. All G-CIMP and IDH1-mut GBMs were characterized by negative  the immune responses [61] doi: 10.1158/0008-5472.CAN-17-3714. ON the other hand, the analyses of complement activation and CD4+, CD8+, or FOXP3+ T-cell infiltration in sections from 72 gliomas of WHO grade III and IV with or without IDH mutations showed significantly reduced complement activation and decreased numbers of tumor-infiltrating CD4+ and CD8+ T cells with comparable FOXP3+/CD4+ ratios. Ex vivo studies demonstrated that 2-HG inhibits complement activation, decreases cellular C3b(iC3b) opsonization and complement-mediated phagocytosis, inhibits T-cell migration, proliferation, and cytokine secretion. This is consistent with reduced host immune responses in IHD-mut gliomas [63]. doi: 10.1158/1078-0432.CCR-17-3855.

CSC/Perictyes –targeting in TME

·         Glioblastoma Stem Cells Generate Vascular Pericytes to Support Vessel Function and Tumor Growth. Lin Cheng,Zhi Huang, Wenchao Zhou, Qiulian Wu, Shannon Donnola, James K. Liu, Xiaoguang Fang,Andrew E. Sloan, Yubin Mao, Justin D. Lathia, Wang Min, Roger E. McLendon, Jeremy N. Rich, and Shideng Bao.

·         Targeting Glioma Stem Cell-derived Pericytes Disrupts the Blood-Tumor Barrier and Improves Chemotherapeutic Efficacy. Wenchao Zhou, Cong Chen, Yu Shi, Qiulian Wu, Ryan C. Gimple, Xiaoguang Fang, Zhi Huang, Kui Zhai,1Susan Q. Ke, Yi-Fang Ping, Hua Feng, Jeremy N. Rich, Jennifer S. Yu, Shideng Bao, andXiu-Wu Bian.

Ad. The mentioned papers are very important, but they do not refer to IDH mutations, so in our opinion  it is unnecessary to include those. We did not intend to cover all issues related to TME.

3. The description of small molecules targeting IDH1/2 should be standardized. i.e. one paragraph describes IC50 values but no others do and reader could compare for self if information standardized. All in vitro findings should be described if available.

Ad. It has been standardized where it was possible to find information about IC50, in a few cases such information is not available.

4. Make sure spaces between each paragraph. DONE.